# Contribution of Autophagy to Cellular Iron Homeostasis and Stress Adaptation in *Alternaria alternata*

**DOI:** 10.3390/ijms25021123

**Published:** 2024-01-17

**Authors:** Pei-Ching Wu, Yen-Ling Choo, Sian-Yong Wei, Jonar I. Yago, Kuang-Ren Chung

**Affiliations:** 1Department of Plant Pathology, College of Agriculture and Natural Resources, National Chung Hsing University, Taichung 402202, Taiwan; jane871057@gmail.com (P.-C.W.); celinechoo95@gmail.com (Y.-L.C.); john90547@gmail.com (S.-Y.W.); 2Plant Science Department, College of Agriculture, Nueva Vizcaya State University, Bayombong 3700, Philippines; jyago2002@yahoo.com

**Keywords:** filamentous fungi, labile iron pool, lipid peroxidation, pathogenicity, ROS, siderophore

## Abstract

The tangerine pathotype of *Alternaria alternata* produces the *Alternaria citri* toxin (ACT), which elicits a host immune response characterized by the increase in harmful reactive oxygen species (ROS) production. ROS detoxification in *A. alternata* relies on the degradation of peroxisomes through autophagy and iron acquisition using siderophores. In this study, we investigated the role of autophagy in regulating siderophore and iron homeostasis in *A. alternata*. Our results showed that autophagy positively influences siderophore production and iron uptake. The *A. alternata* strains deficient in autophagy-related genes 1 and 8 (ΔAa*atg1* and ΔAa*atg8*) could not thrive without iron, and their adaptability to high-iron environments was also reduced. Furthermore, the ability of autophagy-deficient strains to withstand ROS was compromised. Notably, autophagy deficiency significantly reduced the production of dimerumic acid (DMA), a siderophore in *A. alternata*, which may contribute to ROS detoxification. Compared to the wild-type strain, ΔAa*atg8* was defective in cellular iron balances. We also observed iron-induced autophagy and lipid peroxidation in *A. alternata*. To summarize, our study indicates that autophagy and maintaining iron homeostasis are interconnected and contribute to the stress resistance and the virulence of *A. alternata*. These results provide new insights into the complex interplay connecting autophagy, iron metabolism, and fungal pathogenesis in *A. alternata*.

## 1. Introduction

Fungi require iron for their growth and metabolism and have developed diverse strategies to obtain it even though the prevalent form of iron in the environment is insoluble ferric iron (Fe^3+^). The secretion of siderophores is crucial for fungi, as they can bind to ferric iron and transport it via the siderophore-iron transporter (SIT) subfamily into cells [1,2,3,4,5,6]. Fungal siderophores are often produced in multiple types, with hydroxamates being the most common type of siderophores produced by many fungi [7,8]. The intra- and extracellular siderophore biosynthesis shared the same precursor, _L_-ornithine, which is catalyzed by the flavin-dependent monooxygenase siderophore A (SidA) to form *N*^5^-hydroxy-_L_-ornithine [9]. Siderophores are then synthesized by non-ribosomal peptide synthetases (NRPSs) within large enzyme complexes, followed by additional hydroxylation, acylation, and acetylation steps [10]. *Aspergillus fumigatus* produces and secretes various siderophores. Hydroxamates and triacetyl fusarine C (TAFC) are primarily used for external iron uptake, ferricrocin is used for internal iron management, and hydroxyferricrocin is required for conidia germination and oxidative stress resistance [11]. In the phytopathogenic fungus *Cochliobolus heterostrophus*, the absence of Nps6, which is responsible for extracellular coprogen synthesis, results in a two-fold increase in intracellular ferricrocin compared to the wild-type strain [12]. This suggests redirection of the shared precursor towards increased intracellular siderophore production when extracellular siderophore production is impaired. In addition to siderophores, fungi can employ the reductive iron acquisition system (RIA) to acquire iron directly by converting ferric iron to ferrous iron (Fe^2+^) [13]. A recent study has demonstrated that siderophore production and iron acquisition are essential for pathogenicity in the genus *Alternaria* [14]. In the tangerine pathotype of *Alternaria alternata*, Aa*nps6* encoding an NRPS has been shown to play a crucial role in siderophore production and virulence. Strains lacking Aa*nps6* significantly reduce extracellular siderophore production and increase sensitivity to oxidative stress and iron depletion [15,16]. Additionally, AaHapX and AaSreA transcription factors have been found to regulate cellular iron levels by modulating the expression of genes involved in siderophore biosynthesis and iron acquisition [17,18]. These findings underscore the critical role of siderophore-mediated iron uptake in maintaining iron homeostasis, resisting toxic ROS, and ultimately contributing to the virulence of *A. alternata*.

Once iron is taken into the cell via siderophores, it is either released as Fe^3+^ by hydrolyzing the siderophore or converted from the Fe^3+^-siderophore complex into usable Fe^2+^ by ferric reductases [19,20]. Additionally, excess iron can be stored in vacuoles or sequestered by intracellular siderophores, minimizing the “labile iron pool” (LIP) [13,21]. The LIP, rich in Fe^2+^, can cause cellular damage due to the generation of harmful free radicals through Fenton reactions. This can lead to issues like endoplasmic reticulum (ER) stress, mitochondrial dysfunction, and the onset of ferroptosis [22,23]. Ferroptosis, a cell death process associated with iron, is characterized by the occurrence of lipid peroxidation that disrupts membrane stability and integrity, ultimately leading to cell demise [24]. According to recent research, autophagy has been implicated in regulating cellular iron levels [25,26,27]. Autophagy is a cellular process that helps maintain cellular balance during stressful conditions [28,29]. The upregulation of autophagy can alleviate the adverse effects of iron overload within the cell [30]. In mammals, autophagy degrades the iron-storage protein, ferritin, to release iron, and cells can maintain physiological iron balance [31,32]. Autophagy facilitates metal nutrient translocation to promote growth and development in plants [33]. Autophagy also plays a critical role in sustaining microorganism growth under nutrient-scarce conditions by recycling essential metal ions and efficiently utilizing intracellular iron stores [34,35]. Overall, the vital function of autophagy in regulating iron levels highlights its crucial role in ensuring cell survival.

The necrotrophic fungal pathogen *A. alternata* impacts over 100 plant species, utilizing key colonization elements—cell wall-degrading enzymes, appressoria, and the host-selective toxin [36,37]. It employs sophisticated mechanisms for detoxifying ROS, encompassing the mentioned siderophore system, the AaYap1 transcription factor, and the Nox complex (AaNoxA, AaNoxB, AaNoxR) [38,39]. Our previous research also highlighted the significance of autophagy-mediated peroxisome turnover in enhancing cell adaptability under ROS stress [40]. Specifically, the deletion of the Aa*atg8* gene, which encodes the autophagy-related protein 8 (Atg8) crucial for autophagosome biogenesis [41], results in impaired autophagy, reduced virulence, increased ROS sensitivity, peroxisome accumulation, and decreased pathogenicity in *A. alternata* [40]. However, the role of autophagy in regulating siderophore production and intracellular iron homeostasis in *A. alternata* remains unclear. Hence, the present study aims to investigate the role of autophagy, acting as a protective mechanism to alleviate iron stress. Our findings may shed light on potential strategies to combat fungal infections and provide new insights into the intricate relationship between autophagy and iron homeostasis in fungi. Understanding how fungi acquire and regulate iron is crucial for developing new antifungal therapies and preventing *Alternaria* infections.

## 2. Results

### 2.1. AaAtg8 Plays a Role in Iron Uptake during A. alternata Infection

To investigate the relationship connecting virulence, iron uptake, and autophagy, we utilized Prussian blue staining to quantify Fe^3+^ levels in the fungal mycelium grown on the surface of citrus leaves (Figure 1). Prussian blue staining, often used to detect Fe^3+^, could form insoluble and intensely blue granules or deposits when Fe^3+^ reacts with potassium ferrocyanide in an acidic solution [42]. The findings demonstrated the presence of accumulated blue pigment signals within the mycelium of *A. alternata* (wild-type strain) four days after inoculation, signifying the uptake of Fe^3+^ by the wild type during infection (Figure 1). In contrast, the ΔAa*nps6* strain exhibited sporadic blue granules around the mycelium and had lower Fe^3+^ content within hyphae than the wild-type strain. This deficiency likely arose from the ΔAa*nps6* impaired ability to uptake iron, restricting the iron accumulation solely to the periphery of the mycelium without penetration into its interior. Similar staining results were observed in the autophagy-deficient strain ΔAa*atg8*, suggesting the involvement of the autophagic process in iron uptake during infection. Moreover, we found that *A. brassicicola* could acquire iron from the leaf surface, even though this pathogen could not cause disease in citrus leaves.

### 2.2. AaAtg8-Mediated Autophagy Regulates the Production of Intra- and Extracellular Siderophores

We investigated the role of AaAtg8 in iron uptake during infection by examining the ability of the ΔAa*atg8* mutant to produce siderophores on the chromeazurol S (CAS) medium. Our results showed that the ΔAa*atg8* mutant produced significantly smaller orange halos, indicative of siderophore production, around the colony than the wild-type and the complementation strains (Appendix A). We further quantified intra- and extracellular siderophore production using high-performance liquid chromatography (HPLC) analysis. Due to the significantly greater quantity of extracellular siderophores than intracellular siderophores, a 15-fold dilution of the extracellular siderophores was performed before HPLC analysis. The wild type significantly decreased extracellular siderophore production under iron-replete conditions (+Fe, red line) compared to iron-depleted conditions (−Fe, blue line) (Figure 2). Both ΔAa*atg8* and ΔAa*nps6* produced significantly fewer extracellular siderophores than the wild type. In contrast to extracellular siderophores, intracellular siderophores increased in the wild type in response to the presence of iron. Furthermore, ΔAa*nps6* exhibited an increased level of intracellular siderophores compared with the wild type. In contrast to ΔAa*nps6* and wild type, ΔAa*atg8* showed a significant reduction in the production of intracellular siderophores. Liquid chromatography-tandem mass spectrometry (LC-MS/MS) analysis identified various extracellular siderophores, including dimerumic acid, hydroxyneocoprogen, hydroxycoprogen, neocoprogen I, and coprogen. Noticeably, intracellular siderophores contained ferricrocin compared to the extracellular siderophores (Appendix A).

### 2.3. AaAtg8 Influences the Expression of Genes Related to Siderophore Production and Iron Acquisition

Under iron-replete conditions [minimal medium supplemented with 0.5 mM ferric chloride (FeCl_3_), MM+Fe], quantitative real-time PCR (qRT-PCR) analysis revealed that the wild-type strain exhibited significant downregulation of genes, including Aa*nps6*, Aa*sidA*, Aa*sit1*, and Aa*ftr1*, involved in siderophore biosynthesis and iron acquisition (Figure 3). Furthermore, the siderophore positive regulator gene, Aa*hapX*, was downregulated under iron-replete conditions in the wild type. Compared with the wild-type strain, these genes were significantly downregulated in the ΔAa*atg8* mutant under iron-depleted conditions (MM). We also observed upregulation of the siderophore suppressor gene Aa*sreA* in the wild-type strain under iron-replete conditions, which inhibited siderophore production. Interestingly, the ΔAa*atg8* mutant exhibited higher transcriptional levels of Aa*sreA* than the wild-type strain in both iron-replete and iron-depleted conditions.

### 2.4. AaAtg8 Is Required to Maintain Cellular Iron Homeostasis

To investigate the impact of autophagy dysfunction on intracellular iron balance, we employed Prussian blue and FerroOrange to detect the intracellular irons in the wild type and ΔAa*atg8* mutant. Staining fungal hyphae with FerroOrange, a dye tailored to detect Fe^2+^ ions, revealed that ΔAa*atg8* hyphae grown in MM displayed markedly greater fluorescence intensity than those of the wild type (Figure 4). The fluorescence intensity of FerroOrange in the wild type and ΔAa*atg8* decreased upon adding the bathophenanthrolinedisulfonic acid (BPS) to MM. This indicated that autophagy deficiency could lead to the accumulation of Fe^2+^ ions within the cell. The results of Prussian blue staining for Fe^3+^ detection revealed that blue pigments were found in the wild-type mycelium compared to ΔAa*atg8* grown in MM. Remarkably, after 24 h of treatment with the divalent iron chelator BPS, ΔAa*atg8* showed increased intracellular Fe^3+^ iron accumulation, as evidenced by Prussian blue staining, compared to the MM control.

### 2.5. Autophagy Regulates Iron Utilization and Promotes Oxidative Stress Resistance

To explore the role of autophagy in regulating iron utilization and promoting oxidative stress resistance, we assessed the growth of *A. alternata* under different iron concentrations in the presence of oxidative stress. The ΔAa*atg8* strain experienced severe growth impairment under high iron concentrations [MM amended with 0.5 mM ferrous sulfate (FeSO_4_), hFe] and was unable to grow under iron-deficient conditions (MM amended with 0.25 mM BPS), whereas the wild-type strain slightly reduced growth in the presence of hFe or BPS. However, this growth reduction in wild type could be remedied when hFe and BPS were simultaneously added (Figure 5). Similarly, the growth deficit observed in the ΔAa*atg8* strain could be restored using the same approach, although the effect was less pronounced compared to the wild-type strain. Furthermore, the ΔAa*atg8* strain was more sensitive to ROS [MM amended with 15 mM hydrogen peroxide (H_2_O_2_) or 2.5 mM diethyl maleate (DEM)] than the wild type (Figure 5). Under iron-deficient conditions (e.g., in the presence of BPS), the wild-type strain exhibited an impaired ability to withstand oxidative stress. Under H_2_O_2_ stress conditions, adding low-level Fe (25 µM FeSO_4_) restored the growth of both the wild-type and the ΔAa*atg8* strains. However, in contrast to the wild type, the growth impairment of the ΔAa*atg8* strain was only partially restored with the addition of hFe. Noticeably, the addition of high iron concentrations under DEM treatment resulted in a more severe impact on fungal growth, particularly in the ΔAa*atg8* strain. However, adding BPS partially restored fungal growth under DEM/hFe conditions (Figure 5). To confirm the involvement of autophagy-related genes in siderophore production and maintaining iron homeostasis, we examined the autophagy-deficient strain, ΔAa*atg1*. The *atg1* gene encodes an autophagy-related protein 1 (Atg1), which is essential for initiating autophagy [43]. The absence of *atg1* in *A. alternata* (ΔAa*atg1*_D6 and D7) hinders autophagy formation under nutrient deficiency or oxidative stress conditions, thereby impacting the fungi’s pathogenicity on citrus leaves (unpublished data). Similar to ΔAa*atg8*, ΔAa*atg1* reduced siderophore production on CAS plate assays (Appendix A), altered iron utilization, and failed to detoxify oxidative stress effectively (Appendix A). In addition, the complementation strain (CP8), which carries a functional Aa*atg1* gene, exhibited wild-type growth (Appendix A).

### 2.6. Excess Iron Triggers Autophagy and Lipid Peroxidation in A. alternata

To investigate the potential relationship among iron, autophagy, and lipid peroxidation in *A. alternata*, we exposed fungal strains to varying iron concentrations and examined autophagic flux and lipid peroxidation. We assessed autophagy levels using Western blotting by detecting the release of free sGFP from sGFP-AaAtg8 in the wild-type strain, considering our prior in-depth examination of sGFP-AaAtg8 properties in the wild type [40] and anticipating the similarity of sGFP-AaAtg8 in the ΔAa*atg8* strain in both the complementation and wild-type strains. The results revealed that autophagic flux was induced to a greater extent under iron-replete conditions (MM supply FeSO_4_ or FeCl_3_) than under iron-deficient conditions (MM or MM amended BPS) (Figure 6a). We assessed the level of malondialdehyde (MDA), which is a secondary product of lipid oxidation [44,45]. After exposure to high iron concentrations (0.5 mM FeSO_4_), both the wild-type and the ΔAa*atg8* strains exhibited elevated MDA levels, in contrast to conditions of iron deficiency (MM) (Figure 6b).

## 3. Discussion

Fungi have evolved various mechanisms to obtain iron in low-iron environments, and one of the major mechanisms is through the production and secretion of siderophores to acquire iron from a host [46,47]. In this study, we observed that both ΔAa*atg8*, exhibiting defects in autophagy, and ΔAa*nps6*, lacking the ability to synthesize extracellular siderophores, were unable to acquire iron ions from host cells. Notably, both strains showed a significant reduction in virulence. These results indicate that the ability of *A. alternata* to acquire and utilize iron ions from the environment is required for virulence, and autophagy may play a role in mediating the regulation of iron uptake during infection.

Similar to *A. fumigatus*, where siderophore biosynthesis and iron uptake are regulated by the transcription factors SreA and HapX [48,49], *A. alternata* utilizes a comparable regulatory mechanism. AaSreA negatively regulates iron acquisition genes in high-iron environments, and AaHapX activates iron acquisition genes under low-iron conditions [17,18]. Our study demonstrated that autophagy function affects the expression of these genes, thereby influencing siderophore production and transport. Interestingly, the expression of the Aa*sreA* gene significantly increased in the ΔAa*atg8* strain, possibly due to the accumulation of intracellular iron ions. The FerroOrange staining results supported the hypothesis, showing the accumulation of Fe^2+^ within the autophagy-deficient cells. Prussian blue staining of the ΔAa*atg8* strain revealed lower Fe^3+^ content within the cells compared to the wild type in MM, consistent with observations during leaf infections, confirming the impact of autophagy deficiency on Fe^3+^ uptake. Remarkably, after a 24 h treatment with BPS, both the wild type and the ΔAa*atg8* mutant showed increased Fe^3+^ levels. This might be because BPS reduces intracellular Fe^2+^ accumulation, thereby promoting iron uptake. However, BPS supplementation showed limited efficacy under the nutrient and iron-deficient conditions in which the fungi were cultured. Sensitivity experiments further revealed suboptimal fungal growth, especially noticeable in the autophagy-deficient strains in MM+BPS conditions. Even with additional iron supplementation, the growth recovery in the autophagy-deficient strains was less pronounced compared to the wild-type strain. Additionally, the wild type’s ability to detoxify ROS was influenced by iron deficiency. These data underscore the ongoing significance of autophagy in iron utilization for ROS detoxification.

Intracellular siderophores are pivotal for managing iron levels and shielding against the negative effects of iron overload [50]. Fungi must tightly control intracellular siderophore levels, as their absence can impede fungal growth and diminish pathogenic potential [12,13,14]. A deficiency in intracellular siderophore in *A. nidulans* increases the labile Fe^2+^ pool, potentially exacerbating cell oxidative damage, highlighting the interconnectedness of iron metabolism and oxidative stress [51,52,53,54]. Our study has shown that ΔAa*atg8* not only diminished the production of extracellular siderophores but also reduced the production of intracellular siderophores. This influences the stability of intracellular ion balance, particularly the accumulation of Fe^2+^ ions, potentially resulting in oxidative damage to cells. The sensitivity testing results supported this hypothesis, and the ΔAa*atg1* and ΔAa*atg8* strains exhibited significant impairments in resistance to high iron stress. Furthermore, our results indicated that iron ions (Fe^2+^/Fe^3+^) significantly increase the autophagic flux in the wild type compared to the iron depletion conditions (MM or MM+BPS), emphasizing the crucial role of autophagy in detoxifying excess iron.

In addition, the wild-type strain displayed higher resistance to H_2_O_2_ than to DEM under high iron conditions. This could be attributed to the FeSO_4_-mediated decomposition of H_2_O_2_, which partially neutralizes its toxicity [55]. In contrast, failure to neutralize DEM results in the rapid depletion of cellular glutathione (GSH), leading to oxidative stress [56]. Remarkably, HPLC analyses revealed a significant reduction in the presence of the dimerumic acid siderophore in the ΔAa*atg8* strain. Dimerumic acid, a hydrolysis product of coprogen, serves as a potent antioxidant, effectively inhibiting cellular damage by attenuating oxidative stress [57,58]. This suggests a potential mechanism for *A. alternata* to detoxify ROS through siderophores.

In mammals, autophagy is a pro-survival mechanism that regulates iron homeostasis by controlling iron storage proteins such as ferritin [59]. Excessive autophagy can lead to over-digestion of ferritin, increasing iron accumulation and promoting ferroptotic cell death, known as autophagy-dependent ferroptosis [25,60]. However, both ferroptosis and autophagy have interdependent and independent functions in the precise regulation of cell death during pathogenic differentiation [61]. While ferroptosis is generally considered harmful to cells, it has been found to play a vital role in the life cycle of certain organisms. In *Magnaporthe oryzae*, ferroptosis is required for conidial cell death and appressorium maturation [62]. Our studies revealed that iron ions could elevate lipid peroxidation, relevant to ferroptosis, in *A. alternata*. Surprisingly, the absence of autophagy does not appear to impact the occurrence of lipid peroxidation. These findings suggest no clear, direct causal relationship between autophagy and the occurrence of lipid peroxidation in *A. alternata*. However, the impact of lipid peroxidation and the occurrence of ferroptosis remained largely unknown and requires further investigation in *A. alternata*.

## 4. Materials and Methods

### 4.1. Fungal Strains and Culture Conditions

Fungal strains and culture conditions were as follows. The wild-type EV-MIL31 strain of *A. alternata* (Fr.) Keissler has been previously characterized [38]. *A. brassicicola*, which produces host-specific AB toxin and infects plants such as cabbage and cauliflower, was used as a negative control for pathogenicity assays. In addition to utilizing Aa*atg8*-deficient mutants (ΔAa*atg8*_D1 and D2) and a complementation strain (Cp17) re-carrying a functional Aa*atg8* gene in certain experiments, the primary strain employed in this study was ΔAa*atg8*_D1. These strains and the siderophore biosynthesis-deficient strain (ΔAa*nps6*) were previously generated in our laboratory [15,40]. The generation of Aa*atg1*-deficient mutants (ΔAa*atg1*) involved a split marker approach, as outlined in our prior study [63]. Putative fungal transformants (ΔAa*atg1*_D6 and D7) underwent assessment through Southern blot analyses, employing Aa*atg1*- or *hyg*-specific gene probes (Appendix A). A pCB1532 plasmid containing the full-length Aa*atg1* gene and its native promoter region was constructed. This plasmid was then introduced into protoplasts prepared from the ΔAa*atg1*_D6 strain, resulting in the complementation strain (CP8). Comprehensive details of the primers designed for the knockout and variation analysis of the Aa*atg1* gene are listed in Appendix A. The wild-type strain expressing a sGFP-AaAtg8 fusion protein was used to monitor autophagic flux [40]. To produce conidia, all fungal strains were cultured on potato dextrose agar (PDA; Difco, 90000-758, BD, Franklin Lks., NJ, USA) under constant fluorescent light for 3 to 5 days, and conidia were then collected by scraping them off the agar plates with sterile water. For medium shift experiments, conidia grown in potato dextrose broth (PDB; Difco, 90003-494, BD, Franklin Lks., NJ, USA) for 1 to 2 days were washed with sterile Milli-Q water (MQ), and transferred into MM [15] amended with 0.5 mM FeCl_3_, 0.5 mM FeSO_4_, or 0.25 mM BPS. The cultures were incubated for an additional 6 or 24 h.

### 4.2. Sensitivity Assays

To assess the chemical sensitivity of the fungal strains, mycelium was collected using the tip of a 1 mL pipette and transferred onto MM agar medium containing a test compound. The plates were then incubated under constant fluorescent light conditions. The test compounds included 15 mM H_2_O_2_, 2.5 mM DEM, 25 µM or 0.5 mM FeSO_4_, and 0.25 mM BPS. Colony radial growth was measured after 5 days, and each treatment had at least two replicates. The experiments were independently repeated at least three times to ensure the reproducibility of the results.

### 4.3. Virulence Tests

Fungal virulence was evaluated on detached calamondin leaves as previously described [40]. Each leaf spot was inoculated with 10 μL of conidial suspensions containing 10^5^ conidia/mL. Leaf spots treated with sterile MQ were included as a mock control. The treated leaves were then placed in a humid plastic container for 4 to 5 days to develop lesions. The experiment was conducted with at least five leaves for each strain. All tests were repeated a minimum of three times.

### 4.4. Prussian Blue Staining

A histochemical staining technique using Prussian blue was employed to detect Fe^3+^ ions in fungi during infection according to the protocol described by Dangol et al. [64]. Detached calamondin leaves were soaked in a mixture of 7% (*w*/*v*) potassium ferrocyanide and 2% (*v*/*v*) hydrochloric acid (in a 1:1 ratio, *v*/*v*) at room temperature in the dark for 3 days. Afterward, the leaves were kept in a mixture of ethanol, acetic acid, and MQ (in a 92:4:2 ratio, *v*/*v*) for 4 days to ensure discoloration and fixation. In addition, fungal strains cultured in liquid MM or MM supplemented with 0.25 mM BPS for 24 h were subjected to Prussian blue staining without further discoloration or fixation. Fe^3+^ ions reacted with the ferrocyanides in an acid solution to form insoluble and intensely blue pigments.

### 4.5. Analysis of Siderophores

The CAS assay was performed as previously described to evaluate the ability of *A. alternata* strains to produce siderophores [65]. For quantitative analysis of siderophore production, fungal strains were cultured in 100 mL PDB for 2 days at 28 °C with shaking at 150 rpm. The mycelium was then harvested, washed with 100 mL of sterile MQ, and incubated in 300 mL of iron-depleted MM for 5 days. The mycelium was harvested again, divided, transferred to liquid MM or MM supplemented with 0.5 mM FeCl_3_, and incubated for 3 days.

After incubation, the cultures were separated into mycelial and culture filtrate fractions. Culture filtrates were mixed with 1.5 mM FeCl_3_, and extracellular siderophores were purified using Amberlite XAD-16 resin (Sigma-Aldrich, St. Louis, MO, USA) as previously described [66]. To obtain intracellular siderophores, mycelia were washed twice with 50 mL of MQ to minimize extracellular product contamination, ground in liquid nitrogen, suspended in 100 mL of 50 mM KPO_4_, and incubated at room temperature (~25 °C) for 30 min on a shaker. The suspension was filtered, and the filtrate was collected as the intracellular siderophore fraction and purified using Amberlite XAD-16 resin.

The intra- and extracellular siderophores were analyzed using HPLC, following the methods described by Wu et al. [66] with minor modifications. Siderophores were dissolved in a solvent containing 0.2% formic acid in MQ and methanol in a 75:25 ratio. The extracellular siderophores were diluted 15-fold prior to HPLC analysis. The mobile phase used in the experiment comprised a linear gradient of 0.2% formic acid in MQ and 0.2% formic acid in methanol. The composition of the mobile phase changed gradually over time, starting from 75%:25% and shifting to 55%:45% for 10 min, followed by shifting to 25%:75% for 5 min and ultimately reaching 0:100 for the final 10 min at a flow rate of 0.5 mL/min. The type of siderophores was identified via LC-MS/MS analysis using a Dionex UltiMate 3000 UPLC system (Thermo Fisher Scientific, Waltham, MA, USA) equipped with a Phenomenex Luna 5 μm C18 (2) column (250 × 4.60 mm, Phenomenex, Torrance, CA, USA).

### 4.6. Quantitative RT-PCR and Gene Expression Analyses

To measure gene expression levels, qRT-PCR was performed using the method described by Wu et al. [40]. Sequences of gene-specific primers used for qRT-PCR are listed in Appendix A.

### 4.7. Fluorescence Microscopy

Fungal strains were grown in liquid MM or MM amended with 0.25 mM BPS for 24 h. To track intracellular Fe^2+^ levels, hyphae were stained with 10 μM FerroOrang (F374, Dojindo, Kumamoto, Japan) for 30 min in the dark, followed by imaging using a ZOE Fluorescent Cell Imager (Bio-Rad, Hercules, CA, USA) using 543 nm/580 nm filters. All experiments were repeated at least twice.

### 4.8. sGFP-AaAtg8 Proteolysis Assay

To conduct the sGFP-AaAtg8 proteolysis assay, the WT expressing sGFP-AaAtg8 strain was cultured in liquid MM or MM supplemented with 0.25 mM BPS, 0.5 mM FeCl_3_, or FeSO_4_ for 6 h. Crude proteins were extracted and subjected to Western blot analysis using a rabbit polyclonal enhanced GFP (eGFP) antibody (1:5000, kindly provided by Dr. F.J. Jan) to assess the autophagic flux as previously described [40].

### 4.9. MDA Measurement

Lipid peroxidation was assessed by quantifying the level of MDA using the thiobarbituric acid reactive substances (TBARSs) assay [67]. Fungal strains were cultured in liquid MM with or without 0.5 mM FeSO_4_. To prepare the samples, 0.1 g of mycelium was ground to a fine powder with liquid nitrogen and mixed with 0.5 mL of 0.1% (*w*/*v*) trichloroacetic acid (TCA) by vortexing for 15 min. The homogenates were centrifuged at 14,000× *g* for 10 min at 4 °C. The supernatant (150 μL) was transferred to a new tube and mixed with 150 μL of 0.5% TBA (in 20% *v/v* TCA). The mixture was incubated at 80 °C for 30 min, cooled in an ice bath for 5 min, and then centrifuged at 14,000× *g* for 5 min at 4 °C. The absorbance of the resulting TBA-MDA adduct was measured spectrophotometrically using the BioTek Synergy LX Multimode Reader (Agilent, Santa Clara, CA, USA) at wavelength 532 nm, and the non-specific absorbance was subtracted by measuring the absorbance at 600 nm. The MDA concentration was calculated using an extinction coefficient of 155 mM^−1^ cm^−1^ and expressed as nmol MDA per gram of mycelial weight.

### 4.10. Statistical Analysis

All experiments were conducted with a minimum of three replicates unless otherwise stated. Statistical significance was analyzed by performing one-way ANOVA, followed by Tukey’s HSD post-hoc test for multiple comparisons (*p* < 0.05) to determine differences between treatments.

## 5. Conclusions

This study revealed that autophagy facilitates iron uptake and storage in the cell by regulating iron-relative genes and promoting the production of intra- and extracellular siderophores. Autophagy is also involved in iron metabolism in *A. alternata*. The disruption in autophagy significantly weakened the ability to resist ROS and iron stress, further impacting fungal growth, development, and pathogenicity. In addition, we found a potentially protective mechanism involving dimerumic acid siderophore, a potent antioxidant that may be linked to the detoxification of ROS in *A. alternata*. These findings elucidate the important role of autophagy in regulating iron homeostasis and ROS detoxification, which provides new insights and directions for developing novel antifungal strategies. Understanding how fungi acquire and regulate iron can help us develop more selective and potent antifungal treatments, minimizing damage to host cells and laying the foundation for improved therapeutic methods.

## Figures and Tables

**Figure 1 ijms-25-01123-f001:**
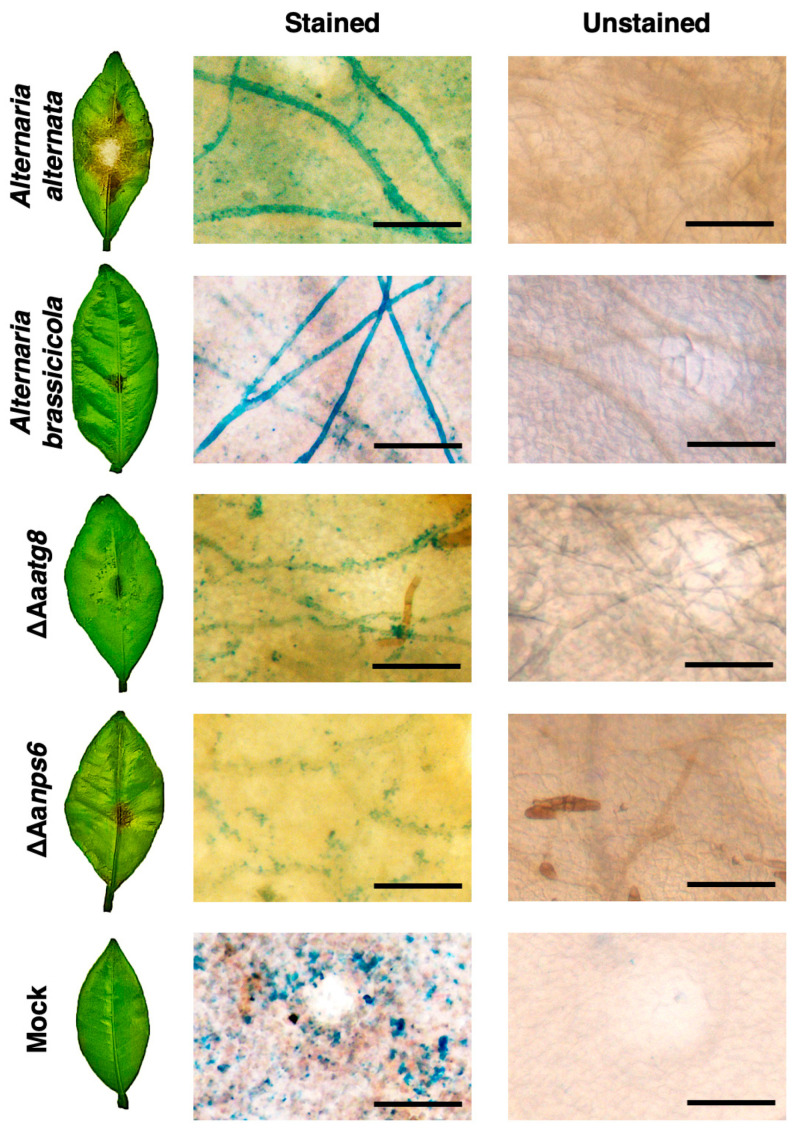
Images of Fe^3+^ accumulation in the *Alternaria* strains inoculated on citrus leaves. Prussian blue staining was used to visualize the accumulation of ferric ions in various fungal strains after being inoculated on detached calamondin leaves. The strains included *A. alternata*, *A. brassicicola*, ΔAa*atg8*, and ΔAa*nps6*. Leaf spots treated with sterile water were used as mock treatment control. After being inoculated for 4 days, the leaves were stained with Prussian blue, and bright blue pigments indicated the accumulation of Fe^3+^. The scale bar in the images represents a length of 50 μm.

**Figure 2 ijms-25-01123-f002:**
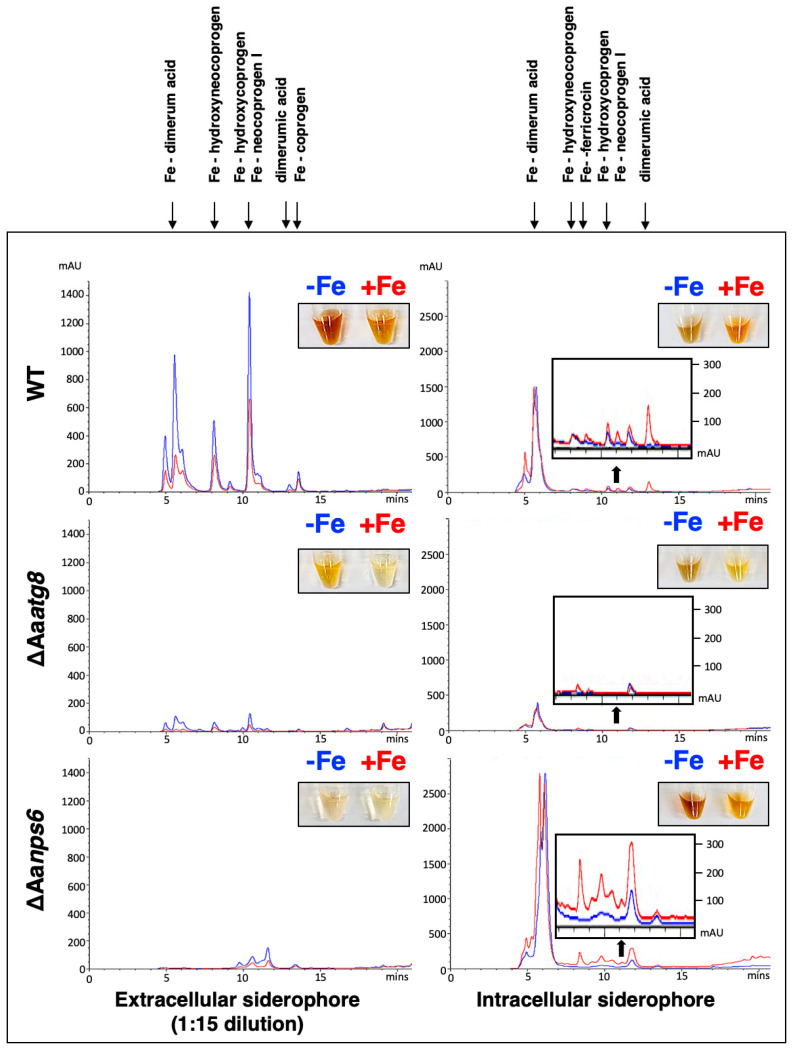
Autophagy is associated with siderophore production. HPLC analysis was performed to measure intra- and extracellular siderophore produced by wild type (WT), ΔAa*atg8*, and ΔAa*nps6* in the presence (+Fe, red line) or absence (−Fe, blue line) of iron conditions. Some HPLC peaks were magnified for better resolution (insets), with the *Y*-axis scale magnified four-fold. The major peaks are labeled with the identities of corresponding siderophores.

**Figure 3 ijms-25-01123-f003:**
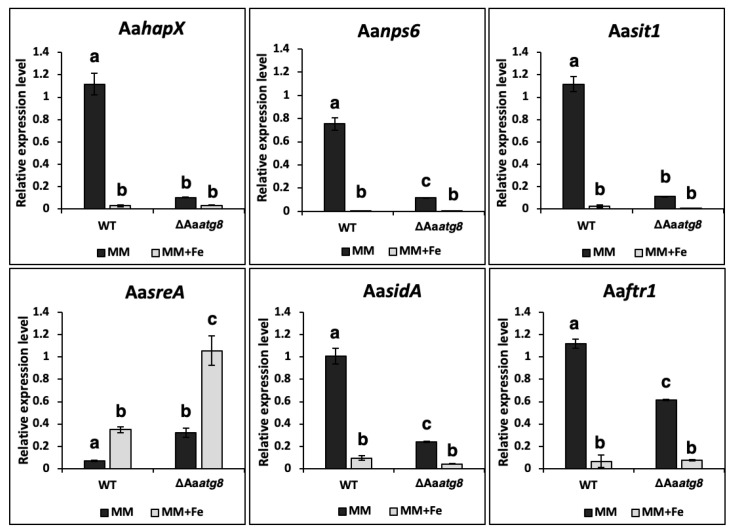
Autophagy influences the expression of siderophore-related genes. The WT and the ΔAa*atg8* strains were cultured in MM or MM amended with 0.5 mM FeCl_3_ (MM + Fe) for 24 h. RNA was isolated from these strains and converted to cDNA. Quantitative RT-PCR was performed using primers targeting genes involved in siderophore production (Aa*nps6* and Aa*sidA*), transporters (Aa*sit1* and Aa*ftr1*), iron regulation (Aa*hapX* and Aa*sreA*), and the β-tubulin-coding gene. Each treatment involved three technical replicates, and the experiments were independently replicated at least twice. The comparative Ct method (ΔΔCT) calculated the relative expression level based on the *β-tubulin* expression level. The data were analyzed for significant differences using one-way ANOVA followed by Tukey’s HSD post-hoc test for multiple comparisons. Means with the same letter indicate no significant difference (*p* < 0.05).

**Figure 4 ijms-25-01123-f004:**
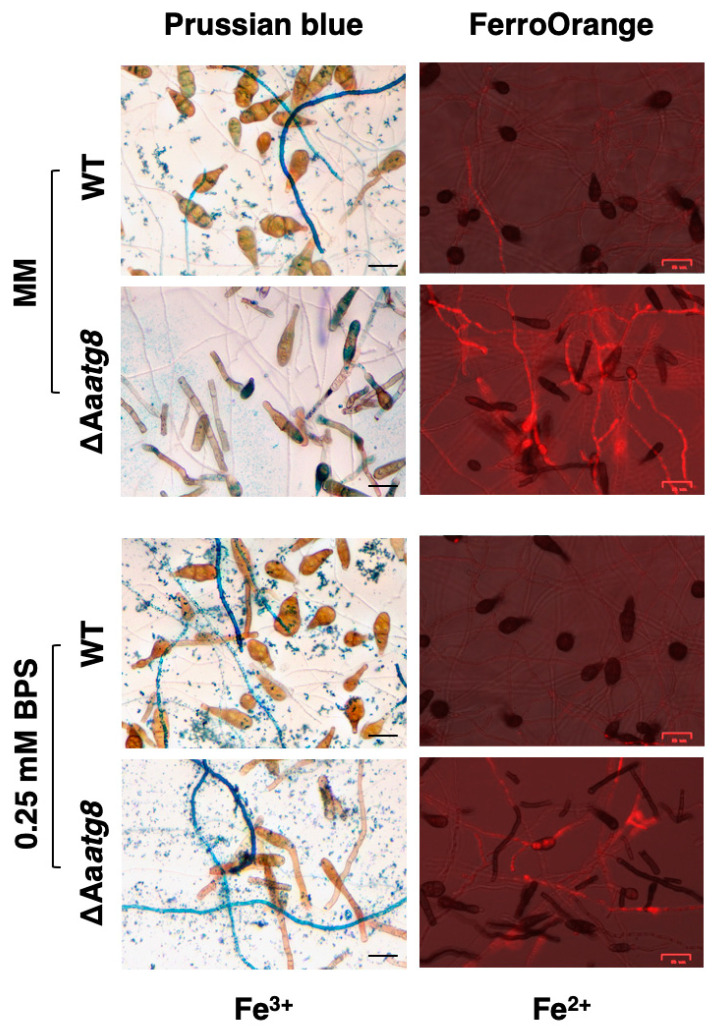
Autophagy impacts intracellular ion homeostasis. Prussian blue and FerroOrange staining were used to assess intracellular iron levels in both the WT and ΔAa*atg8* strains, cultivated in either MM or MM supplemented with 0.25 mM BPS for 24 h. Prussian blue staining detected Fe^3+^, showing blue granules or deposits. FerroOrange produced a red fluorescence upon binding to Fe^2+^. The scale bar in the images represents a length of 25 μm.

**Figure 5 ijms-25-01123-f005:**
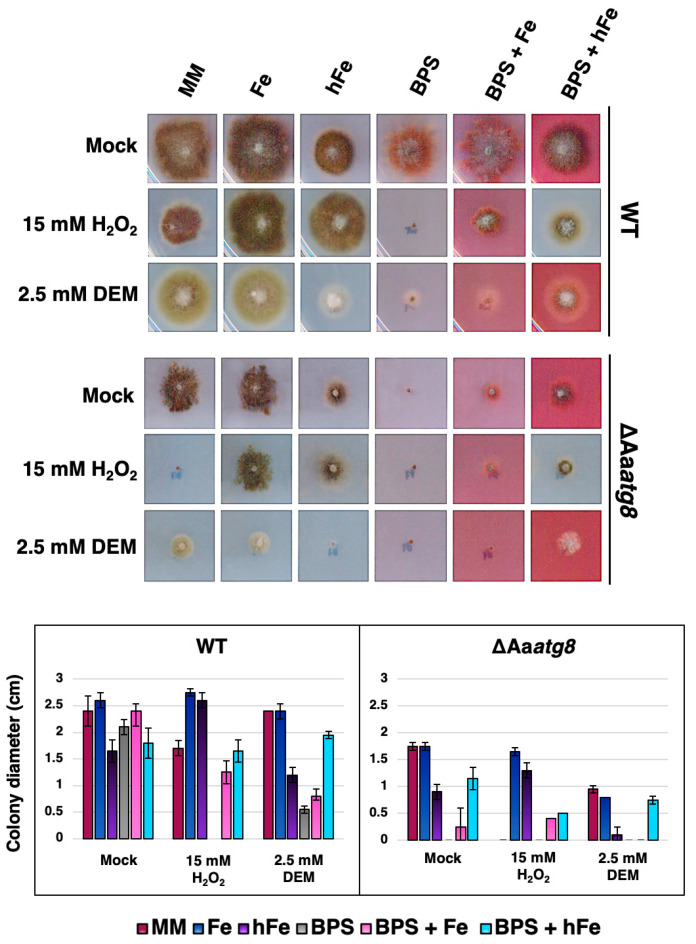
Autophagy promotes iron utilization and impacts resistance to oxidative stress. The WT and the ΔAa*atg8* strains were grown on MM or MM supplemented with 25 μM FeSO_4_ (Fe), 0.5 mM FeSO_4_ (hFe), 0.25 mM BPS (the iron chelator), 15 mM H_2_O_2_, or 2.5 mM DEM (the oxidative stress inducer). Colony diameters were measured and recorded after 5 days of incubation at 28 °C. Each treatment had a minimum of two replicates, and the experiments were repeated at least three times to ensure result reproducibility. The presented results are the means ± standard deviations from three independent experiments.

**Figure 6 ijms-25-01123-f006:**
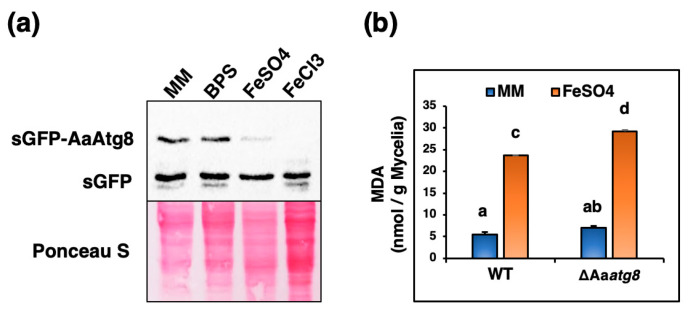
Correlations among iron, autophagy, and lipid peroxidation in *A. alternata.* (**a**) Western blot analysis was performed to detect free sGFP released from sGFP-AaAtg8, indicative of the occurrence of autophagy in WT after being treated with iron-depleted (MM, MM with 0.25 mM BPS) and replete conditions (MM with 0.5 mM FeCl_3_ or FeSO_4_) for 6 h. Ponceau S stain was used to check sample loading equality. (**b**) The MDA concentrations were measured in the WT and the ΔAa*atg8* treated with or without 0.5 mM FeSO_4_ for 6 h. Means with the same letter indicate no significant difference.

## Data Availability

The data presented in this study are available from the corresponding author upon reasonable request.

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
