# Peer review of "Contribution of Autophagy to Cellular Iron Homeostasis and Stress Adaptation in Alternaria alternata"

_ijms, 2024, doi:10.3390/ijms25021123_

Round 1
Reviewer 1 Report
Comments and Suggestions for Authors
Wu et al. studied about contribution of autophagy to iron homeostasis and stress tolerance in Alternaria alternata. Manuscript was well written. Eventhough, authors needs to address following comments before it get accepted in IJMS.
Comment 1: If possible, change title.
Comment 2: Ln.10, please write abbreviation for ACT.
Comment 3: Ln 16. write abbreviation for ΔAaatg1 and ΔAaatg8
Comment 4: Ln. 21-24, rewrite the sentences.
Comment 5: Ln. 51, change A. alternata to Alternaria alternata
Comment 6: Lot of abbreviations in the text, please check throughout the manuscript, make it easy for readers.
Comment 7: In Introduction section, i didnt find much information about Alternaria alternata and its autophagy and iron homeostasis and stress adaptation. Please include more information.
Comment 8: Figure 1, write full scientific name of the organism
Comment 9: Figure 3, write number of replicates and standard deviation or standard errors.
Comment 10: ln. 212, you mentioned two replicates with two experiments, most of the scientific experiments will be carried out three or more, two replicates will affect your results, justify.
Comment 11: If possible, improve discussion
Comment 12: If possible, include separate section for conclusion. Please check typographical errors throughout the manuscript.
Reviewer 2 Report
Comments and Suggestions for Authors
Autophagy plays an important role in the fungal pathogenesis of plants. The study of the molecular mechanisms involved between plants and pathogenic fungi will aid in the understanding of autophagic processes and fungal pathogenicity. The current study focused on the effects of an autophagy-deficient Alternaria alternata ΔAaatg8 strain on siderophore production and iron uptake to know the relationships between iron homeostasis, ROS detoxification, and fungal pathogenesis. The results indicated that autophagy affected the expression of iron homeostasis-related genes and showed resistance to oxidative stress. The connections between iron, autophagy, and lipid peroxidation were also investigated. The manuscript seems to provide solid evidence, but the essential information is lacking. It’s not recommended for the International Journal of Molecular Sciences in its current form. The major reasons are listed below.
1. There is no supplementary material to evaluate, for example, the data from another autophagy-deficient strain ΔAaatg1.
2. The importance of studying Atg1 and Atg8 in relation to autophagy was ignored. Only a previous report on the Aaatg8 gene impaired autophagy, reduced virulence, and increased ROS sensitivity and peroxisome accumulation was mentioned. The roles of Atg1 and Atg8 related to pathogenicity in fungi and autophagic processes should be described.
3. The characteristics of the autophagy-deficient strain, ΔAaatg1, were not described or cited.
4. Two Aaatg8-deficient mutants (ΔAaatg8_D1 and D2), a complementation strain (Cp) re-carrying a functional Aaatg8 316 gene, two Aaatg1-deficient mutants (ΔAaatg1_D6 and D7), and a complementation strain (CP8), re-carrying a functional Aaatg1 gene, were mentioned in Section 4.1. Which Aaatg8-deficient mutant and Aaatg1-deficient mutant were used in this study? In addition, two complementation strains didn’t show further investigations in this study, except those shown in the supplementary materials.
5. The scale of the x- and y-axis in Figure 2 was hard to read. It affected the interpretation of the results. For example, extracellular siderophores may contain ferricrocin though it was shown. The magnification of the insets of intracellular siderophores should be indicated. The question that did the peaks after magnification reveal statistical significance should be answered.
6. Why the western blot analysis was not performed to detect free sGFP released from sGFP-AaAtg8 in ΔAaatg8 strain or ΔAaatg1 strain after being treated with iron-depleted and replete conditions? At least the data of ΔAaatg8 strain, the main mutant in this study, should be shown for comparison with that of WT.
7. The strategy of understanding how fungi acquire and regulate iron is crucial for developing new antifungal therapies and preventing Alternaria infections could be discussed.
Round 2
Reviewer 2 Report
Comments and Suggestions for Authors
Autophagy plays an essential role in the fungal pathogenesis of plants. Studying the molecular mechanisms involved between plants and pathogenic fungi will aid in understanding autophagic processes and fungal pathogenicity. The report focused on the effects of an autophagy-deficient Alternaria alternata ΔAaatg8 strain on siderophore production and iron uptake to know the relationships between iron homeostasis, ROS detoxification, and fungal pathogenesis. The data from another autophagy-deficient strain, ΔAaatg1 was also included. The current study indicated that autophagy affected the expression of iron homeostasis-related genes and showed resistance to oxidative stress. The connections between iron, autophagy, and lipid peroxidation were also investigated. The revised manuscript was improved to provide the required information to support the relationships between autophagy and iron homeostasis. It is suggested that the following improvements be made before recommendation to the International Journal of Molecular Sciences.
1. In addition to the A. alternata ΔAaatg8 strain, several pieces of evidence were supported by characteristics of another autophagy-deficient strain, ΔAaatg1. The molecular procedures, the construction, and the validation of ΔAaatg1_D6 and D7 and a complementation strain (CP8), re-carrying a functional Aaatg1 gene, should be described in detail. Since they were not found in the previous study, they must be shown in Section 4.1 and the supplementary materials as supporting information in the authors’ previous publication (doi: 10.1111/mpp.13247).
2. Section 4.1. It was described that a complementation strain (Cp) re-carrying a functional Aaatg8 gene in specific experiments, the primary strain employed was ΔAaatg8_D1. It was uncertain if the complementation strain (Cp) was the Cp17 strain, which was also described in the authors’ previous publication (doi: 10.1111/mpp.13247). It has to be mentioned it’s the Cp17 strain or another complementation strain.
3. Similarly, two Aaatg1-deficient mutants (ΔAaatg1_D6 and D7) and a complementation strain (CP8), re-carrying a functional Aaatg1 gene, were mentioned in Section 4.1. However, another complementation strain (CP18) was used for the experiments. In addition, the primary strain employed carrying a functional Aaatg1 gene was not described.
4. Figure 2. Several identities corresponding to different types of siderophores labeled above the respective peaks were missing. They include dimerumic acid and coprogen for extracellular siderophores and dimerumic acid, hydroxyneocoprogen, hydroxycoprogen, neocoprogen I, and coprogen for intracellular siderophores.
5. It is suggested to indicate the scale of the Y-axis in Figure 2 insets to show their possible biological meanings, though the Y-axis scale is magnified fourfold as described.
6. The western blot analysis was not performed to detect free sGFP released from sGFP-AaAtg8 in ΔAaatg8 strain or ΔAaatg1 strain after being treated with iron-depleted and repleted conditions. Though introducing sGFP-AaAtg8 into the ΔAaatg8 mutant is conceptually similar to the complementation strain, where the characteristics are restored to wild-type (WT) levels, it’s a speculation and needs evidence. At least, it is suggested to describe the speculation, based on the authors’ opinion, in Section 2.6 to explain why the data of ΔAaatg8 strain, the primary mutant in this study, was not shown for comparison with that of WT.
Round 3
Reviewer 2 Report
Comments and Suggestions for Authors
Autophagy plays an essential role in the fungal pathogenesis of plants. Studying the molecular mechanisms between plants and pathogenic fungi will aid in understanding autophagic processes and fungal pathogenicity. The report focused on the effects of an autophagy-deficient Alternaria alternata ΔAaatg8 strain on siderophore production and iron uptake to know the relationships between iron homeostasis, ROS detoxification, and fungal pathogenesis. The data from another autophagy-deficient strain, ΔAaatg1 was also included. The current study indicated that autophagy affected the expression of iron homeostasis-related genes and showed resistance to oxidative stress. The connections between iron, autophagy, and lipid peroxidation were also investigated. The revised manuscript addressed all the questions and doubts. It was also improved to provide the required information to support the relationships between autophagy and iron homeostasis. After careful evaluation, it is recommended to the International Journal of Molecular Sciences.